# Modeling 2020 regulatory changes in international shipping emissions helps explain 2023 anomalous warming

Ilaria Quaglia[1] and Daniele Visioni[2]

[1]Sibley School of Mechanical and Aerospace Engineering, Cornell University, Ithaca, NY
[2]Department of Earth and Atmospheric Sciences, Cornell University, Ithaca, NY

**Correspondence:** Daniele Visioni (dv224@cornell.edu)

**Abstract.** The summer of 2023 has seen an anomalous increase in temperatures even when considering the ongoing greenhouse-gases driven warming trend. Here we demonstrate that regulatory changes to sulfate emissions from international shipping routes, which resulted in a significant reduction in sulfate particulate released during international shipping starting on January 1 2020, have been a major contributing factor to the monthly surface temperature anomalies during the last year. We do this by including in Community Earth System Model (CESM2) simulations the appropriate changes to emission databases developed for the Climate Model Intercomparison Project version 6 (CMIP6). The aerosol termination effect simulated by the updated CESM2 simulations of +0.14 ± 0.07 W/m$^2$ and 0.08K ± 0.03K is consistent with observations of both radiative forcing and surface temperature, manifesting a similar delay as the one observed in observational datasets between the implementation of the emission changes and the anomalous increase in warming. Our findings highlight the importance of considering realistic near-future changes in short-lived climate forcers for future climate projections, such as for CMIP7, for an improved understanding and communication of short-term climatic changes.

## 1 Introduction

In 2020, the International Maritime Organization (IMO) prescribed changes in the sulfur content of shipping fuels which resulted in a strong reduction in the particulate emissions of the sector, especially over the North Atlantic corridor. Past research indicated that such change would lead to a minor increase in the global Earth Energy Imbalance (Partanen et al., 2013), mainly through a reduction in cloud formation (Jin et al., 2018). To date, no Coupled Model Intercomparison Project Phase 6 (CMIP6) Earth System Model simulations have directly tried to ascertain the magnitude and role that this abrupt change in forcing would have on the Earth System, which we try to do here in the Community Earth System Model (CESM). We leverage the CESM2 Large Ensemble (LENS2, Rodgers et al. (2021)), which post-2014 uses an emission pathway (the Shared Socio-economic Pathway (SSP) 3-7.0) that does not consider any change in shipping emissions post-2020, and consider a new ensemble of simulations with the only change starting in January 2020 with a sudden drop in sulfate shipping emissions of 90%, hereby named NOSHIP or CESM2-LENS2 without shipping emissions. This drop is consistent with changes reported in (Hoesly and Smith, 2024) (see Methods and Fig. A1). In fact, none of the existing scenarios used in CMIP6 (Meinshausen et al., 2020) include substantial changes in shipping emissions in the present or near future of the same magnitude as those planned by

25 the IMO. Our simulations can help bridge this gap and provide a useful reference point for future emission scenarios such as those for the Phase 7 of CMIP (Meinshausen et al., 2023). The relevance of air quality measures on regional climate has been highlighted before (Zheng et al., 2020; Takemura, 2020; Schumacher et al., 2024): however, the IMO change case may present a unique case study in which global radiative balance may have been measurably affected due to a sharp, almost-instantaneous, event, thereby providing a useful benchmark for climate models' sensitivity.

## 2 Results

### 2.1 Top of Atmosphere radiative forcing

Compared to the CESM2-LENS2 in the default emission scenario, our simulations show an increase in Absorbed Solar Radiation (ASR) and net radiation at the top of atmosphere (TOA, longwave+shortwave) that is more consistent with the increase as measured by CERES (Fig. 1); furthermore, this overall increase is consistently observed as coming from an increase in Cloud
Radiative Forcing (CRF, see Fig. A2), and not from a direct aerosol forcing (Clear Sky) as in CERES (Fig. A3). This is also confirmed by the different magnitude of optical depth changes simulated by CESM2 (Fig. A2). The global NET RF effect as diagnosed by the change between LENS2 and NOSHIP (Fig. 1c, orange and green lines, respectively), is 0.14 W/m$^2$, within the range of previous estimates (with a minimum of +0.06 to a maximum of +0.37 W/m$^2$ across different studies (Partanen et al., 2013; Yuan et al., 2024; Yoshioka et al., 2024; Skeie et al., 2024; Forster et al., 2024)). This indicates that CESM can better
capture short-term forcing changes if the proper emission changes are accounted for, and being more consistent with actual anthropogenic aerosol emissions would help reconcile CESM projected TOA energy imbalance with the one observed. It is worth highlighting that ERA5 reanalyses also present a considerable mismatch between their radiative fluxes and the observed ones, posing question about the reliability of this reanalyses product for this specific purpose.

Our results are also consistent with previous studies in which aerosol emissions have already been shown to present a source of bias for CESM2 compared to observations (Ramachandran et al., 2020; Zhang et al., 2019) in other regions.

### 2.2 Temperature impacts

It has already been highlighted that global surface temperatures in 2023 cannot be explained by natural variability alone Rantanen and Laaksonen (2024). In Fig. 2 (see also Fig. A5-A7) we try to understand if such observed temperature anomalies
can be connected with the shipping emission changes. Even if a radiative forcing change at a specific time can be detected, its translation to a temperature signal can be complicated by natural variability and internal climatic oscillations, and by time-lags due to slow oceanic response and cloud adjustments: as an example, estimates of the exact magnitude of the cooling signal from Pinatubo have varied (0.14-0.5 K, Canty et al. (2013)) even if the forcing change has been robustly evaluated Soden et al. (2002); Schmidt et al. (2018).

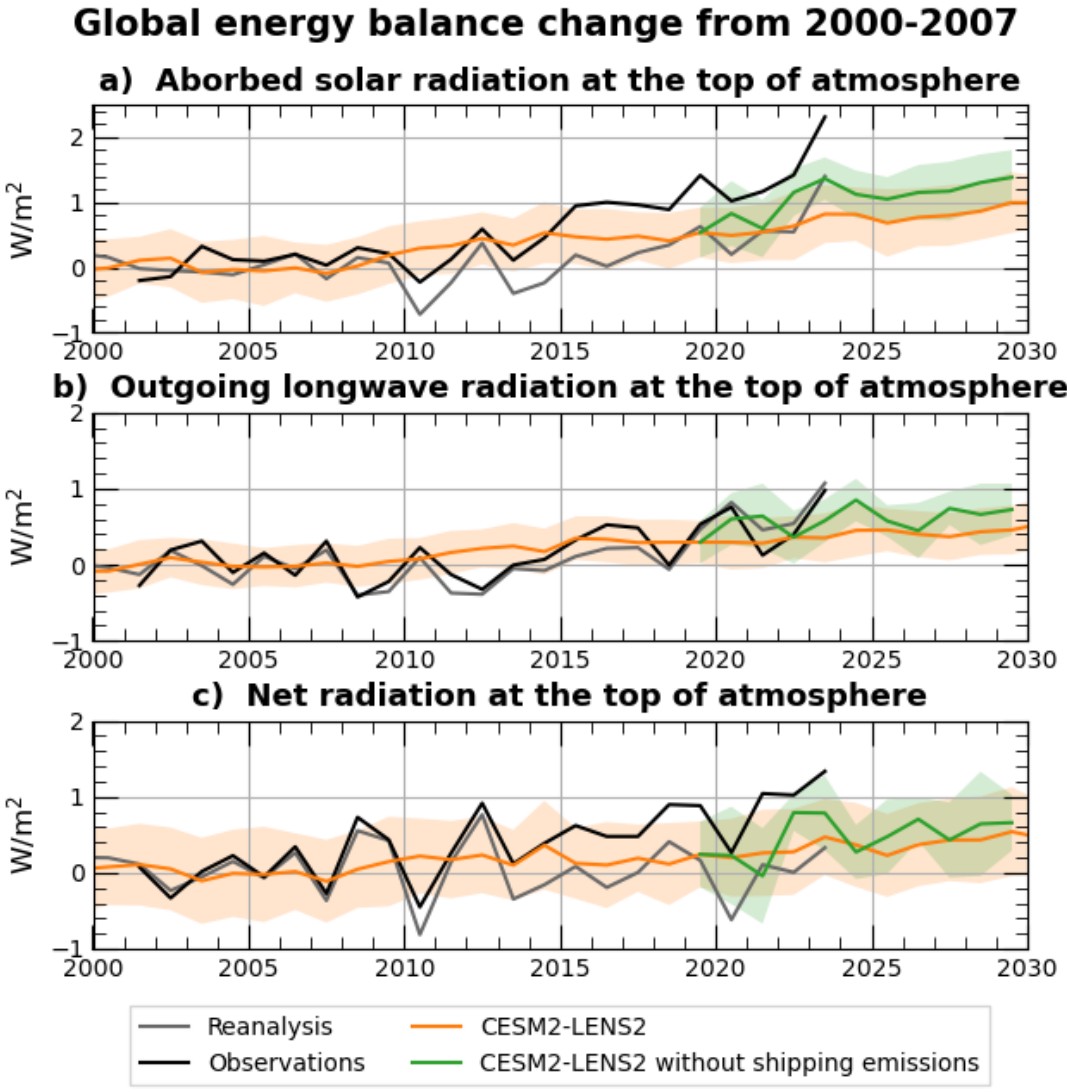

**Figure 1.** Time series of annual mean deviation from the 2000-2007 period for globally averaged (a) Absorbed Solar Radiation (ASR, defined as incoming minus outgoing shortwave), (b) Outgoing Longwave Radiation (OLR) and (c) ASR minus OLR (NET) radiative flux at the top of the atmosphere (TOA). The black line represents observational data from CERES, the gray line reanalysis from ERA5, the orange and green lines the ensemble mean of CESM2-LENS2 simulations with and without shipping emissions, respectively. The shaded area for CESM2-LENS2 simulations represents one standard deviation calculated on ensemble members.

Here we use our two ensembles to perform an attribution of the 2023 temperature impact by comparing their results in the three years following the change in emission between the two scenarios. We compared our model's results with the anomalies in various surface air temperatures datasets (see Fig. A4), but considering the small variations across them, here we only show the model's results compared to Berkley (see Methods). We looked at monthly de-trended global temperature anomalies over the period 2020-2023, i.e. removing the, assumed linear, contribution from greenhouse gasses and seasonality. Figure 2b shows

the monthly de-trending method applied to August months which has been used for each month of the 3-year time series in Figure 2a. For the NOSHIP ensemble we considered an average and high estimates of shipping emission impact (light and dark green lines, respectively), calculated as the ensemble average and ensemble average plus one standard deviation in NOSHIP. The likelihood of occurring temperatures in Figure 2c and d is calculated for the measured temperature anomalies (black line) and for the measured temperature anomalies without the average and high contribution estimated from ship emissions (light

and dark red lines corresponding to the light and dark green lines of panel 2a, respectively).

In LENS2, the mean ensemble response does not show anomalous increases in any of the years under analyses, whereas the NOSHIP ensemble demonstrates a striking agreement with observations in the manifestation of an anomaly compared to past years starting in June 2023, so 3 years after the change in shipping emission (Fig. 2d). Analysis of the ENSO state (Fig. A8) suggests that the anomaly cannot solely be attributed to a strong El Niño event starting in 2023, as it appears in

ensemble members even during El Niña states. This indicates that CESM responds to the global mean increase in incoming solar radiation in a way that is physically consistent with observations, manifesting in a delay in the surface temperature response with a time-lag of the order of three years.

Compared with the overall distribution of expected monthly anomalies in LENS2 (Fig. 2c), measured anomalies from Berkeley are statistically unlikely for the months from July to December (more than 2 $\sigma$), with September peaking with a likelihood

of 0.02 %; considering the temperature contribution from shipping emissions, our results indicate that the same anomalies were roughly 9 times more likely to occur, and 27 times if we consider the higher-end estimate of the shipping contribution estimated by CESM (average + 1 $\sigma$ of the NOSHIP ensemble) (Fig. 2d). For each month but September, this result indicates that, on its own, changes in shipping emission can be considered as a primary contributor capable of explaining the 2023 anomalies in light of internal variability.

The changes observed in the global mean also do not necessarily translate to an even change across the globe: in Fig. 3a we also show which regions, based on the difference between the two CESM ensembles, show significant changes in their annual mean temperatures in 2023. Our results indicate that most of the regions affected are ocean ones, especially in the North and South Atlantic and over the Pacific, with areas of significance over land in the Middle East and at high latitudes in the Northern Hemisphere. This is explainable by observing where most of the changes in absorbed solar radiation at the surface are, which

we show in Fig. 3b considering the integrated change over the 2021-2023 period. In Fig. 3c and 3d we show that the changes in shipping emissions can also have some detectable effects on daily temperature extremes. Using the $90^{th}$ percentile of daily maximum temperature (see Methods), we show that, while the variability is quite high, a signal is distinguishable especially in late 2023 and in 2024 in the NOSHIP ensemble, with a higher percentage of very hot days especially in 2024.

## 3 Conclusions

Our results underscore the importance of considering realistic changes in aerosol emissions when discussing the evolution of surface temperatures at different timescales: we show both that, by 2030, the overall temperature increase is projected to be 0.14 K (Figure A4), and that in the 2020-2024 period there is clearly a heightened signal that contributes to the anomalous, sudden increase. Compared to previous research analyzing aerosol emission changes that mostly happened over land (i.e. Forster et al. (2020) in the case of COVID), in this case the complex interplay of cloud-mediated changes, absorbed solar radiation

by the ocean, and ocean response may help explain why the emergence timescale is different. Our findings also points to the fact that future policy decisions around abrupt reductions in tropospheric aerosols might want to take into account their surface temperature impact: while clear that policies that improve air qualities save lives Partanen et al. (2013), the presence of recent international agreements such as the Paris Agreement focused on avoiding future increases in temperature indicate that such air-quality focused policies, and more broadly other possible human and natural emission changes, may also need to be evalu-

ated in the context of breaches of global temperature thresholds and related potential damages and risks. For instance, it would be legitimate to ask if such policies should more explicitly be framed in terms of estimates of the remaining carbon budget before they're enacted (Rogelj et al., 2019).

Our forcing estimates of +0.14 W/m$^2$ ± 0.07 from CESM2 is located within the range of other recent works, which often

used different methodologies to come to their conclusions. For instance, Yuan et al. (2024) found a forcing of 0.2 ± 0.11 W/m$^2$ (over the global oceans) indirectly estimating it from cloud changes as simulated by NASA's Global Earth Observing System, which would result in a global forcing very close to ours overall. However, their temperature estimate of 0.16K is twice as large as our estimate of 0.08K ± 0.03. Other studies like Skeie et al. (2024) tried to estimate the effective radiative forcing by conducting fixed-SSTs similar to ours, using four models (CESM2-CAM6, NASA GISS ModelE, NorESM2, OsloCTM3), and

finding a range of 0.06 to 0.09 W/m$^2$, similar to Yoshioka et al. (2024) which found 0.13 W/m$^2$ using HadGEM3-GC3.1, very close to the 0.14 W/m$^2$ ± 0.02 W/m$^2$ found in UKESM by Jordan and Henry (2024) under similar experimental protocols. In this latter case, they also estimate the temperature response in UKESM to be 0.046K ± 0.010K. A future assessment of the different methodologies used will be necessary to reconcile these estimates, perhaps coupling it with a rigorous multi-model assessment in CMIP7. A cause for our estimate being towards the higher end of others might be our use of fully coupled

simulations, which may result in a warming-driven feedback on cloud forcing, and which we pursued to try to reconcile our estimates of both forcing and temperature changes with available observations. It is also possible that our results are overestimated due to an excessive sensitivity of CESM2 to cloud-aerosol interaction, or that they are driven by our ensemble size, and our specific ways of detecting 'significance' (for instance, see Watson-Parris et al. (2024) for a different interpretation of this while using a very similar modeling set-up as ours): if this is the case, however, it will be necessary to find other explanations

for the 2023 anomalous temperatures that currently don't seem to exist - especially as the persistent anomaly even after 12 months appears to rule out a statistical fluctuation.

Finally, we note that Forster et al. (2024) suggested that the 2024 global aerosol radiative forcing was made more negative due to the contribution from the Canadian wildfire. While our study is not suited to directly quantify such a potential contribution, due to prescribed biomass burning emissions that predate that specific event, we note that our analyses of CERES fluxes show a global positive increase in Clear sky fluxes between 2023 and 2024 (Fig. A2) that is hard to reconcile with such a hypothesis. While it is certainly possible that increased warming resulted in higher wildfire risks at high latitudes (as suggested by our Fig. 3a results), satellite data of top-of-atmosphere radiative fluxes does not seem to support a forcing compensation between sulfate from shipping and wildfire aerosols. However, future studies including also realistic biomass burning could better clarify such matters.

Future analyses will shine further light on both the reliability of CESM's response in terms of changes in cloud cover and boundary layer sulfate and in terms of the response of such changes on the regional and global radiative forcing and temperature response. It is clear from this work already that in many cases the CESM ensemble is already underestimating the increasing radiative imbalance and, for instance, the near-term projection of max daily temperatures. However, here we think we have successfully argued that further analyses of the current years' changes should not ignore real-world policy changes, both those already happened and future planned ones.

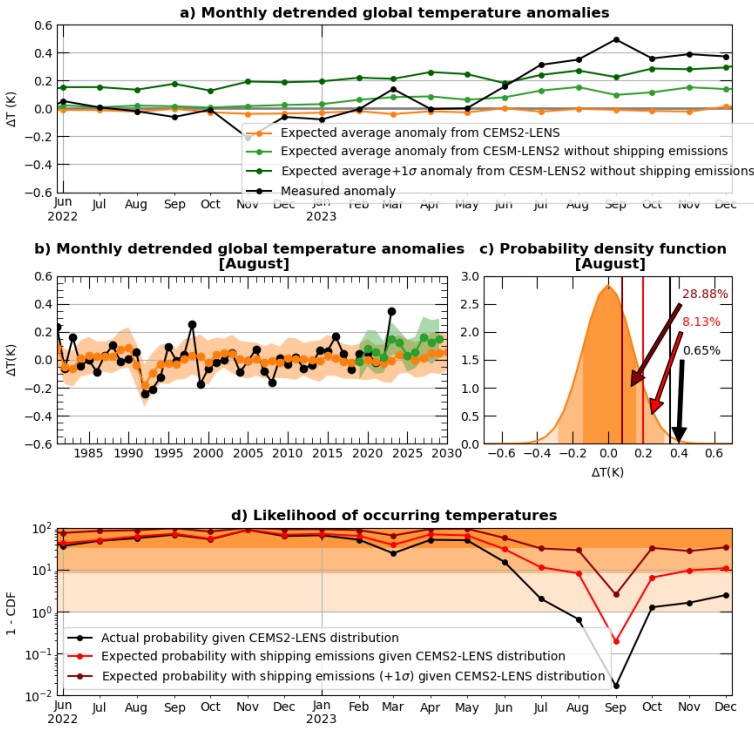

**Figure 2.** a) Detrended deseasonalized values of monthly mean global temperature changes from the 1981-2000 average. The black line represents observational data from Berkeley (in the legend: measured anomalies), the orange and light green line represent the ensemble mean of CESM2-LENS2 simulations with and without shipping emissions (in the legend: average anomaly from CESM2-LENS2 with and without shipping emissions, respectively), and the dark green line is the sum of the ensemble mean plus one standard deviation ($1\sigma$) of CESM2-LENS2 simulations without shipping emissions (in the legend: expected average + $1\sigma$ anomaly from CESM2-LENS2 without shipping emissions). b) Time series of globally averaged detrended temperature change from 1981-2000 as in (a) but only for the month of August, for observational data (black line), the ensemble mean of CESM2-LENS2 simulations with shipping emissions (orange line) and the ensemble mean of CESM2-LENS2 simulations without shipping emissions (light green line), as defined in the legend of panel (a). The shaded area represents one standard deviation on the ensemble members. c) Probability density function (PDF) of globally averaged detrended temperature change from 1981-2000 for the month of August in CESM2-LENS2 with shipping emissions, corresponding to the values of the orange line shown in panel (b). The PDF includes values from 1981 to 2020 for all ensemble members. Vertical lines represent the year 2023 for the measured anomalies (black line), anomalies due to shipping emissions (red line, calculated as difference between observed anomalies and average ensemble anomalies from CESM2-LENS2 without shipping emissions), and average ensemble anomalies plus one standard due to shipping emissions (dark red line, calculated as difference between observed anomalies and average ensemble anomalies plus one standard deviation from CESM2-LENS2 without shipping emissions). The values (in %) represent the right-tail values with respect to the CESM2-LENS2 PDF (one minus the cumulative density function, 1- CDF) for the three vertical lines. Legend is shown in the following panel. The three shaded area, from darkest to lightest orange, represent values within $1\sigma$, $2\sigma$ and $3\sigma$. d) Time series of right-tail values as defined in panel (c) for the month of August in 2023, here for each month of the year from June 2022 to the end of 2023.

**Changes due to shipping emission reduction**

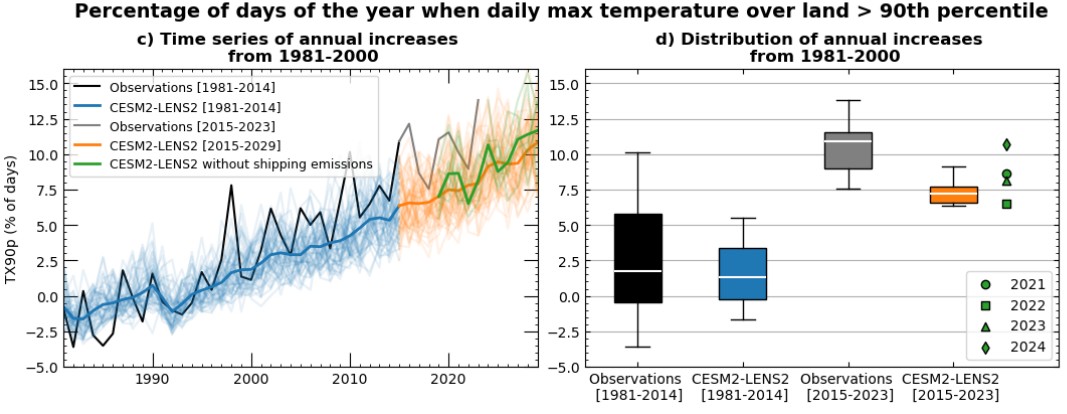

**Figure 3.** Maps of changes in surface air temperature in 2023 (a) and cumulative absorbed solar radiation over 2021-2023 (b) due to reduction in shipping emissions in CESM2-LENS2. Shaded areas indicate regions where the differences are not statistically significant at the 10% level, green contours indicate regions where the differences are not statistically significant at the 5% level. c) Time series of annual increases of percentage of days of the year when daily maximum temperature over land are greater than the $90^{th}$ (TX90p) from 1981-2000 for observations from Berkeley and CESM2-LENS2, distinguished in pre- and post-2014, and CESM2-LENS2 without shipping emissions. d) Box plots of TX90p shown in panel c for Berkeley and ensemble-mean values in CESM2-LENS2, distinguished in pre- and post-2014. Green markers represent years from 2023 to 2024 for ensemble-mean values in CESM2-LENS2 without shipping emissions.

*Data availability.* Simulation data used in this work are available at Quaglia (2024).

## Appendix A: Methods

### A1    CEMS2-LENS2 simulations

The Large Ensemble Community Project (LENS2) consists of 100-member ensemble simulations that cover the historical period 1850-2014 (referred as Historical) and the period 2015-2100 under the SSP3-7.0 emission scenario (referred as SSP3-7.0). The simulations are performed Rodgers et al. (2021) with the Community Earth System Model version 2 run within the Community Atmosphere Model version 6 (CESM2-CAM6) Danabasoglu et al. (2020).

Ensemble members are initialized to capture to capture the internal variability of the climate system: ten members are initialized from different years of the pre-industrial control cycle, from 1001 to 1191 to minimize drift, and forty are initialized from four initial dates (1231, 1251, 1281, and 1301) that identify the phases of the Atlantic Meridional Overturning Circulation (AMOC), and for each starting date the ensemble members were created by randomly perturbing the atmospheric potential temperature field. We used 50 of the 100 ensemble members, i.e., members with smoothed biomass burning emissions of the CMIP6 protocol (11-year running means over the period 1990-2020 (Rodgers et al., 2021).

We additionally performed 10-member ensemble simulations under the SSP3-7.0 scenario (referred as NOSHIP or CESM2-LENS2 without shipping emissions), branched from the available ensemble members with start date 1011-1191 and smoothed biomass burning emissions of the CMIP6 protocol, reducing sulfur emissions from shipping Hoesly et al. (2018) by 90% everywhere over the oceans in accordance with IMO 2020 regulations from 2020 to 2030 (Fig. A1a) and keeping all other emissions the same. The drop in sulfur emissions results in an average reduction over this decade compared to the reference period 2000-2007 of 4.2 Tg-S/yr globally. Due to our computational constraints, we only performed 10 ensemble members compared to the original ensemble of 50. However, following Tebaldi et al. (2021) and Frankcombe et al. (2018), our ensemble size is representative of the perturbed state to allow us to estimate forcing changes.

### A2    Observations and reanalysis

Sulfur shipping emissions are from the Community Emission Data System (CEDS) which provides estimates of emissions of anthropogenic greenhouse gases, reactive gases and aerosols, from 1750 to nowadays, based on existing emission inventories, emission factors, and activity/driver data (Hoesly and Smith, 2024).

To compare simulated radiative fluxes, we used satellite data from the Clouds and the Earth's Radiant Energy System (CERES) Energy Balanced and Filled Top-of-Atmosphere fluxes version 4.2 (CERES_EBAF_Edition4.2, NASA/LARC/S-D/ASDC (2023)) and climate reanalysis data from the fifth generation European Centre for Medium-range Weather Forecasts (ECMWF) reanalysis (ERA5, Hersbach et al.).

Simulated surface temperatures are compared with ERA5, Berkeley Earth, Met Office Hadley Centre/Climatic Research Unit global surface temperature data set version 5.0.2.0 (HadCRUT5), and the NOAA Global Surface Temperature Dataset

version 6.0 (NOAAGlobalTempv6). For the results in our Figure 2 we tested all datasets but ultimately only showed Berkley, as our conclusions were largely independent of the dataset chosen (see Figure A5).

Berkeley Earth Land/Ocean Temperature Record (Rohde and Hausfather, 2020) combines the Berkeley Earth land-surface temperature field with an interpolated version of the Met Office Hadley Centre Sea Surface Temperature dataset version 4.0.0.0 (HadSST4). HadCRUT5 (Morice et al., 2021) uses a statistical infilling method to integrate sea-surface temperature data from the HadSST4 with land-surface air temperature data from the Climatic Research Unit temperature dataset version 5.0.0.0 (CRUTEM5). NOAAGlobalTempv6 (Huang et al.) combines the land-ocean surface temperature analysis from the Extended Reconstructed Sea Surface Temperature (ERSSTv5) with land surface air temperature analysis, which are is on the Global Historical Climatology Network-Monthly (GHCN-M) temperature database.

## A3    Calculation of temperature anomalies

Temperature anomalies for both observations and modeling results are calculated by subtracting the average value on the 1981-2000 period. Each month is detrended considering the timeseries for each month separately, removing the trend calculate as a linear fit on the 1981-2020 period.

## A4    Probability density function of detrended temperature anomalies

The probability density function (PDF) for each month of the year (Fig. A6) is calculated on the detrended temperature anomalies from 1981 to 2020 (Fig. A5), as define before, in CESM2-LENS2 with shipping emissions simulation, including all ensemble members, henceforth referred as CESM2-LENS2 PDF. The right-tail values are calculated as one minus the cumulative density function (1 - CDF) given CEMS2-LENS PDF for observational data from Berkeley, and the so called "Expected probability with shipping emissions given CEMS2-LENS distribution" and "Expected probability with shipping emissions $+1\sigma$ given CEMS2-LENS distribution" (see Fig. 2c and d).

The latter two are the right-tail values, given the CEMS2-LENS distribution, for the detrended temperature anomalies calculated as the difference in the temperature anomalies minus the linear trend in the period 1981-2020 between observed values and the ensemble average in CESM2-NOSHIP ("expected probability with shipping emissions given CEMS2-LENS distribution") and between observed values and the ensemble average plus one standard deviation in CESM2-NOSHIP ("expected probability with shipping emissions $+1\sigma$ given CEMS2-LENS distribution"). This allows us to calculate the probability that the monthly temperature anomaly would have happened if the further warming contribution from shipping emissions had not been removed, treating the LENS2 as the counterfactual world where shipping emissions are maintained, and the difference between the observational dataset and the NOSHIP mean as the counterfactual observational dataset in a world that had maintained shipping emissions unaltered.

## A5    Calculation of Extreme Temperature Index

TX90p is a climate change index defined as the percentage of days when daily maximum temperature greater than the $90^{th}$ percentile of daily maximum temperature (TX90p), according to CCl/CLIVAR/JCOMM Expert Team (ET) on Climate Change Detection and Indices (ETCCDI, http://etccdi.pacificclimate.org). We use this as a measure of changes in extreme temperatures over land that might be relevant for future impacts assessment.

Thresholds for TX90p are calculate as described in Zhang et al. (2005) but without using a bootstrap procedure: the $90^{th}$ percentile of daily maximum temperature is calculated for each day of the year over a base-period of 40 years, from 1961 to 2010 (and including all ensemble-members, in CESM case), and using data from 5 consecutive days centred on the day of interest. This means that in CESM the threshold for a given day of the year is calculated over a distribution of 40x50x5 data (years x ensemble-members x days). Each calendar day of the time series is compared with the respective threshold for that day of the year. We validated our calculations for CESM by applying the same method to the Berekley data set and comparing our results with those calculated by Climpact (https://www.climdex.org/, with results shown in Fig. A9 where we demonstrate a close match between available results and ours).

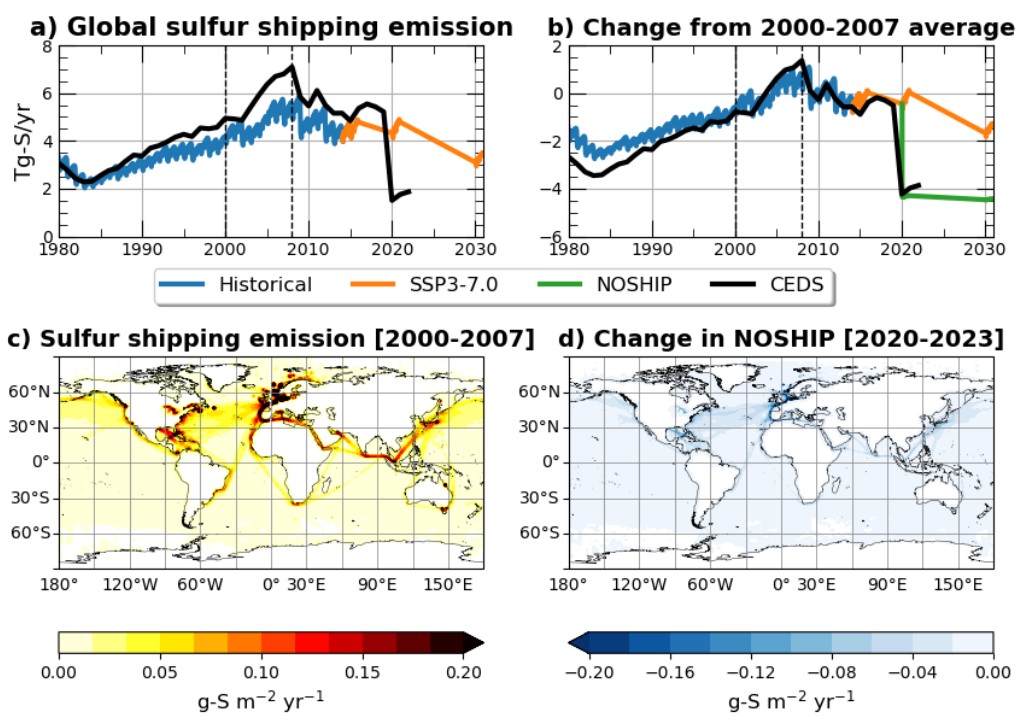

**Figure A1.** a, b) Time series of the global mean of sulfur ($SO_2$ + $SO_4$, in Tg-S/yr) shipping emissions and its change from the from the 2000-2007 period in CESM2-LENS2 and Community Emission Data System (CEDS). Dotted vertical lines define the reference period 2000-2007. c, d) Maps of sulfur shipping emissions averaged over the reference period and the change in sulfur shipping emissions (g-S/yr/m$^2$) from the reference period averaged over the years 2020-2023, respectively.

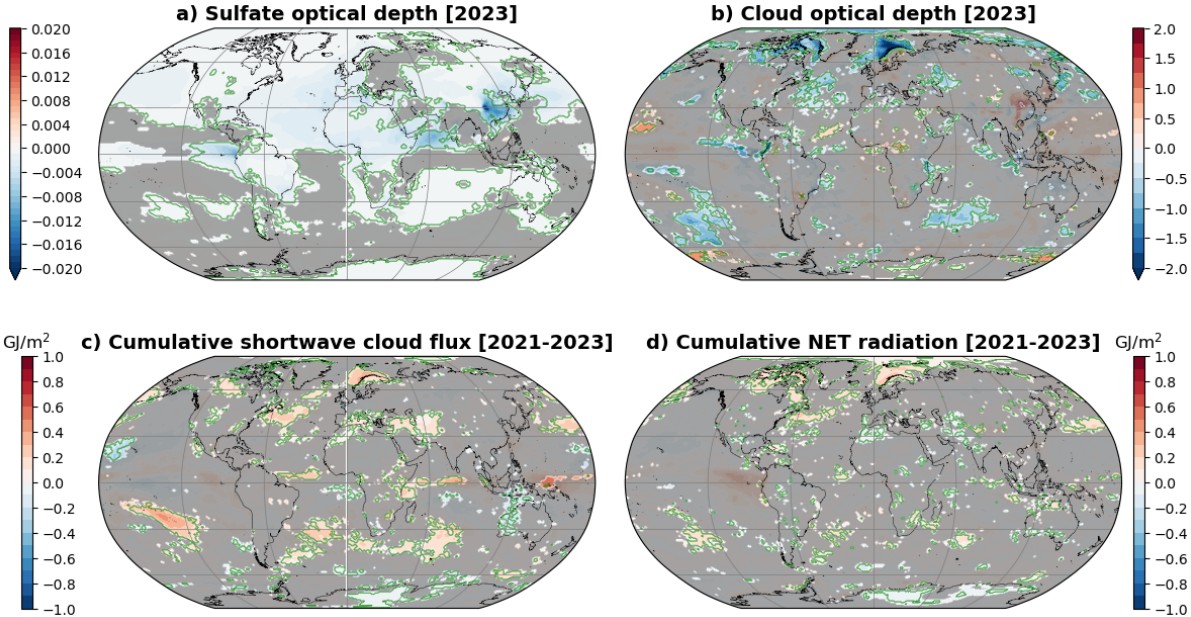

**Figure A2.** Maps of changes in sulfate and cloud visible optical depth in 2023 (a and b, respectively), and cumulative shortwave cloud flux and NET radiation over 2021-2023 (c and d, respectively) due to reduction in shipping emissions in CESM2-LENS2. Shaded areas indicate regions where the differences are not statistically significant at the 10% level, green contours indicate regions where the differences are not statistically significant at the 5% level.

## Global fluxes change from 2000-2007 - Clear Sky

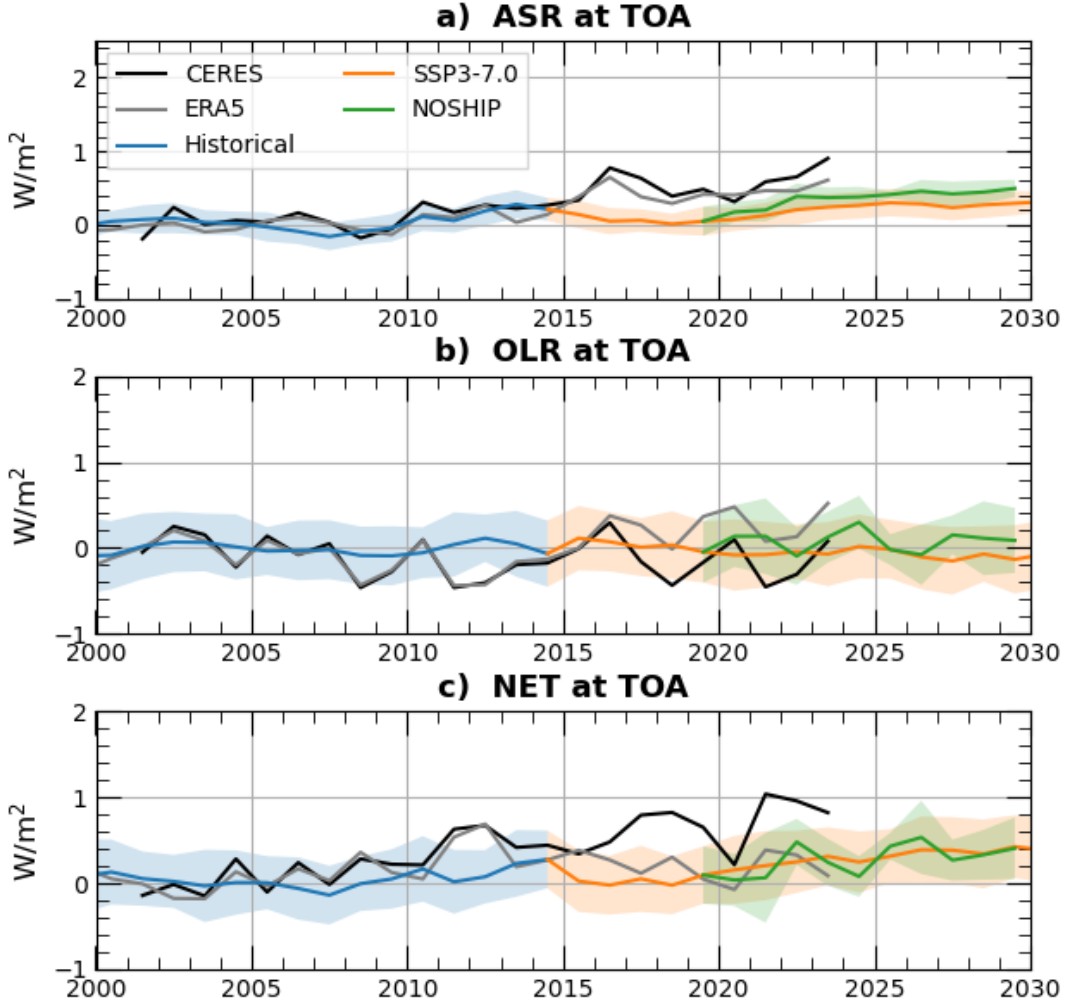

**Figure A3.** Time series of annual mean deviation from the 2000-2007 period for globally averaged (a) Absorbed Solar Radiation (ASR, defined as incoming minus outgoing shortwave), (b) Outgoing Longwave Radiation (OLR) and (c) ASR minus OLR (NET) radiative flux at the top of the atmosphere (TOA), in clear sky conditions. The shaded area for CESM2-LENS2 simulations (Historical, SSP3-7.0, NOSHIP) represents one standard deviation calculated on ensemble members.

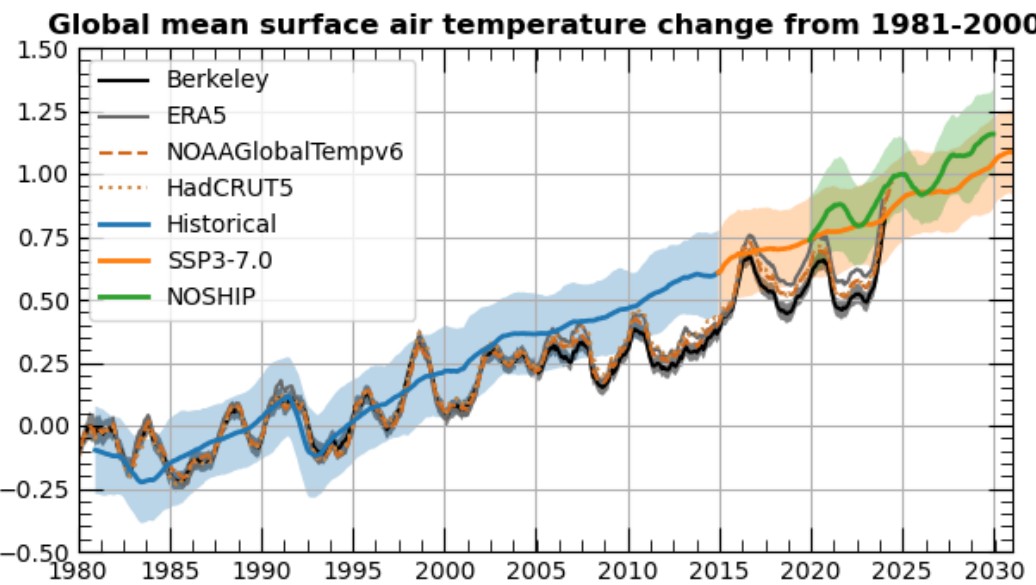

**Figure A4.** Time series of 12-months rolling mean of global mean temperature changes from 1981-2000 for CESM2-LENS2 simulations (Historical, SSP3-7.0, NOSHIP) and observations and reanalysis (Berkeley, ERA5, NOAAGlobalTempv6, HadCRUT5). The shaded area for CESM2-LENS2 simulations represents one standard deviation calculated on ensemble members.

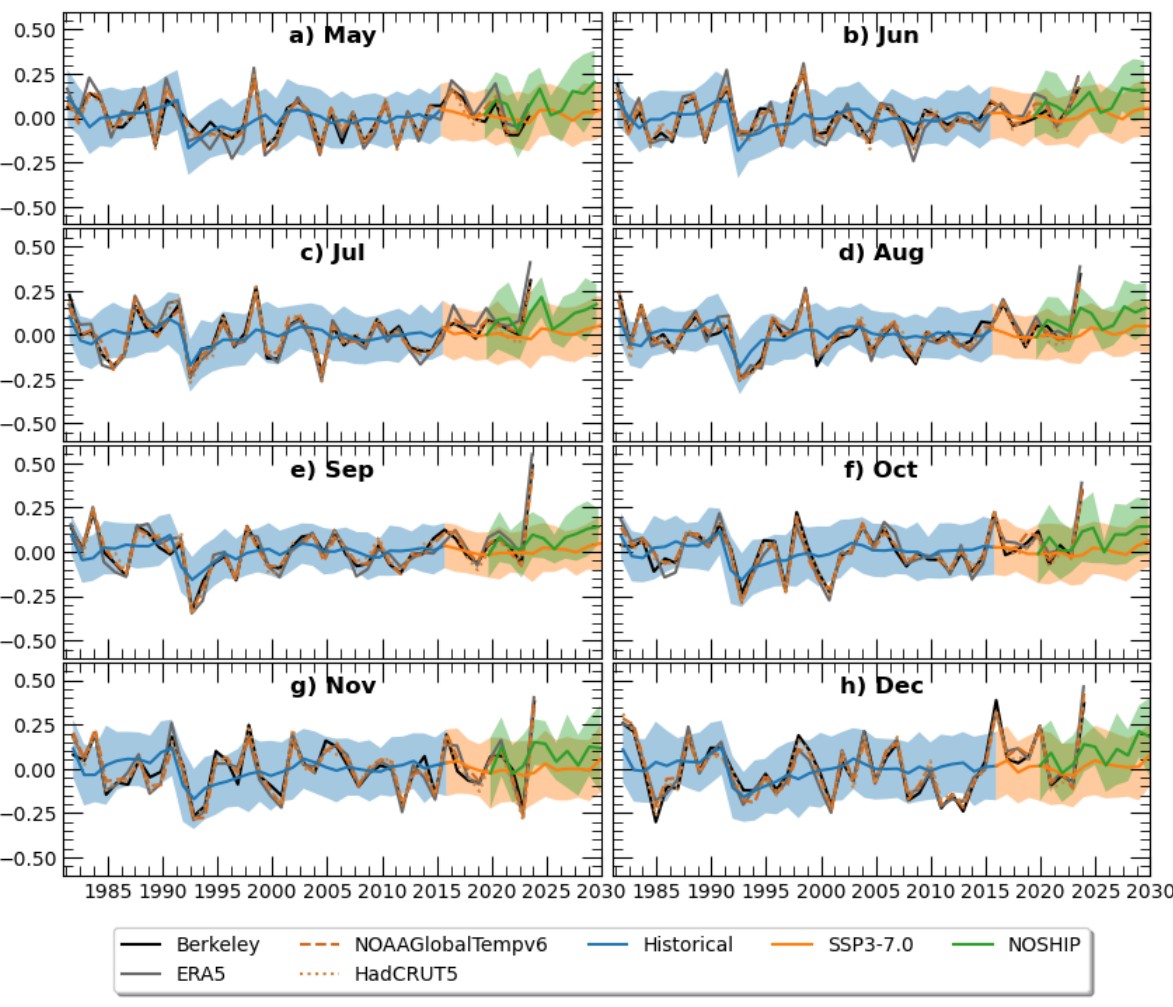

**Figure A5.** Time series of detrended global mean temperature changes from the 1981-2000, distinguished for each month of the year, for CESM2-LENS2 simulations (Historical, SSP3-7.0, NOSHIP) and observations and reanalysis (Berkeley, ERA5, NOAAGlobalTempv5, HadCRUT5). The shaded area for CESM2-LENS2 simulations represents one standard deviation calculated on ensemble members.

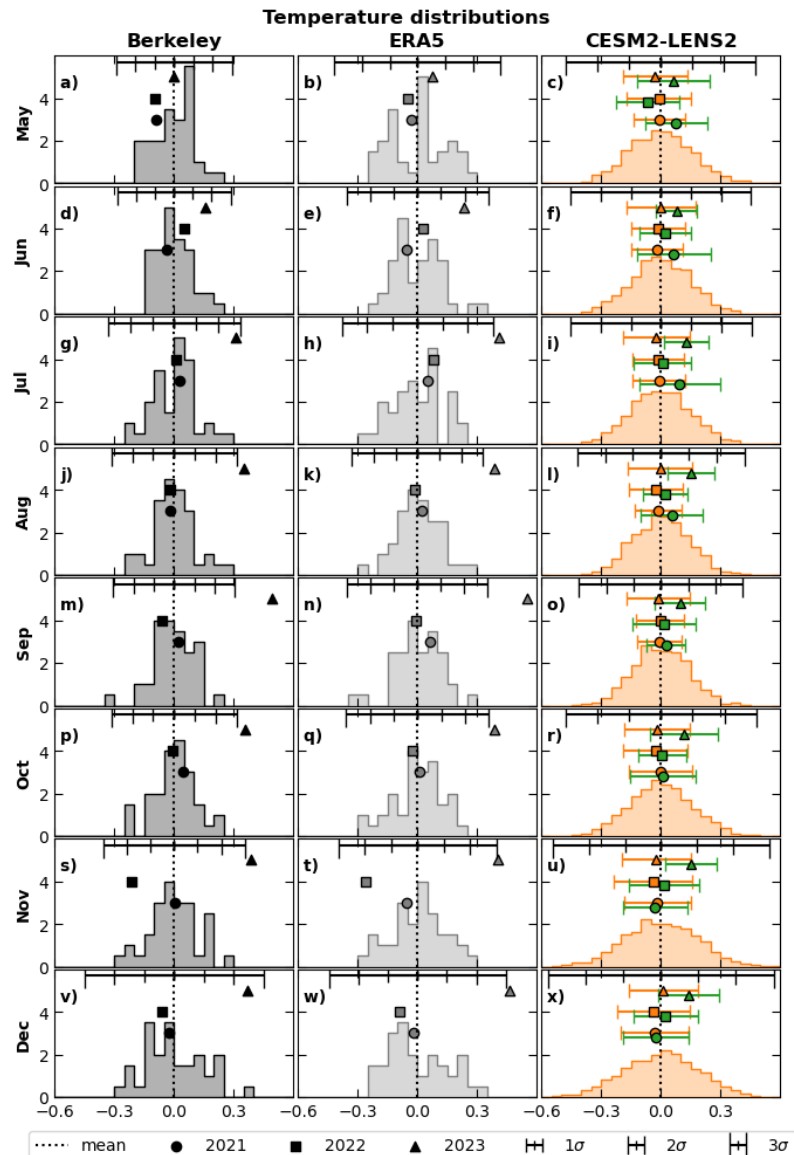

**Figure A6.** Probability density function of temperature changes from the 1981-2000 mean for values from 1981 to 2020 in Berkeley (first column), ERA5 (second column), and CESM2-LENS2 (Historical+SSP2-4.5, third column). Values from 2021-2023 are shown separately with different markers as defined in the legend. The dashed vertical line represents the mean value and the error bars 1σ, 2σ, and 3σ calculated from the respective PDF 1981-2020. CESM2-LENS2 panels includes all ensemble members, the 2021-2023 values for both SSP3-7.0 (orange) and NOSHIP (green) simulations, and the standard deviation calculated on the ensemble members.

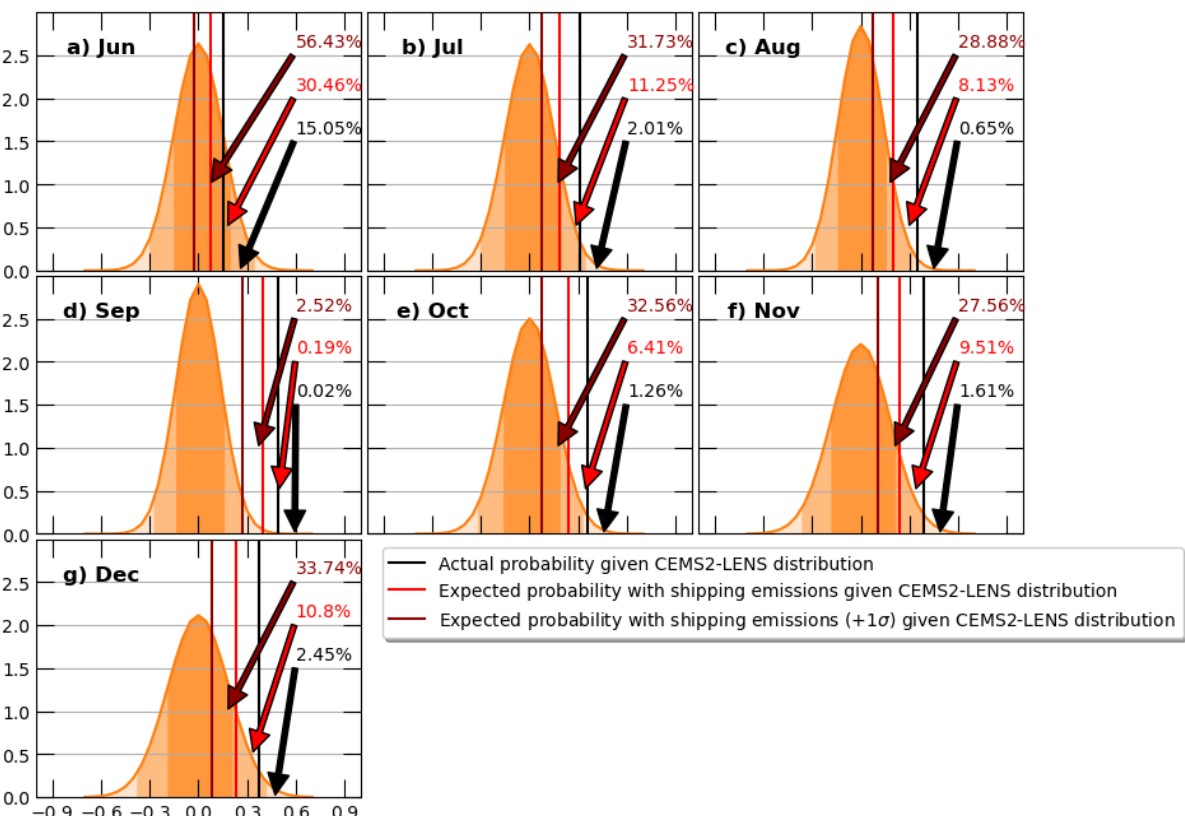

**Figure A7.** Probability density function (PDF) of globally averaged detrended temperature change from 1981-2000 in CESM2-LENS2 for the months of the year from May to December. The PDF includes values from 1981 to 2020 for all ensemble members. Vertical lines represent the year 2023 for the measured anomalies (Berkeley, black line), anomalies due to shipping emissions (red line, calculated as difference between Berkeley and average ensemble anomalies from NOSHIP), and average ensemble anomalies plus one standard due to shipping emissions (dark red line, calculated as difference between Berkeley and average ensemble anomalies plus one standard deviation from NOSHIP). The values (in %) represent the right-tail values with respect to the CESM2-LENS2 PDF (one minus the cumulative density function, 1- CDF) for the three vertical lines. The three shaded area, from darkest to lightest orange, represent values within $1\sigma$, $2\sigma$ and $3\sigma$.

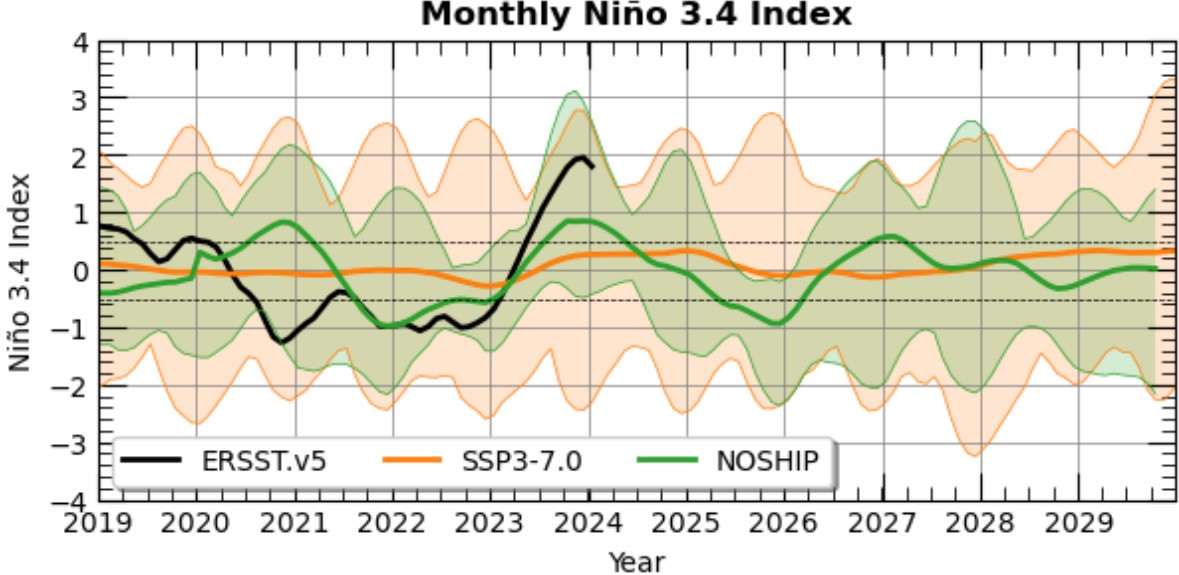

**Figure A8.** The El-Niño 3.4 anomalies. In CESM2-LENS2, the index is calculated using a 20-yr sliding climatology (1950-1999 from Historical) and smoothing the anomalies with a 5-month running mean. The shaded area for CESM2-LENS2 simulations (SSP3-7.0 and NOSHIP) represents one standard deviation calculated on ensemble members. ERSST.v5 data are taken from https://www.ncei.noaa.gov/access/monitoring/enso/sst.

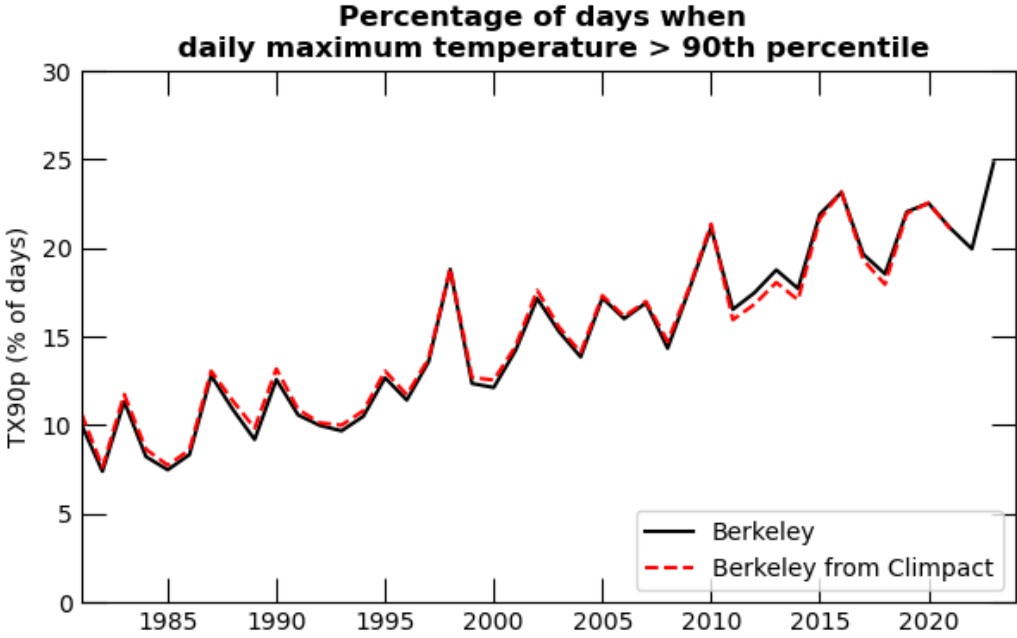

**Figure A9.** Percentage of days of the year when daily maximum temperature greater than the $90^{th}$ percentile of daily maximum temperature (TX90p) from the Berkeley data set. The black line comes from our calculations (see Methods section), compared with Climpact data (dashed red line) taken from https://www.climdex.org/.

*Author contributions.*  IQ performed the CESM2 simulations, analysed the results and assisted with the writing of the manuscript. DV conceptualized the study, wrote the manuscript and assisted with analyses.

*Competing interests.*  The contact author has declared that none of the authors has any competing interests.

*Acknowledgements.*  The authors would like to thank the Cornell Atkinson Center for Sustainability, that made this research possible through
a Fast Grant Award to DV.

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
