# Peer review of "Modeling 2020 regulatory changes in international shipping emissions helps explain 2023 anomalous warming"

_EGUsphere, 2024_

## Referee Comment (RC1)

This paper presents a useful contribution to the attribution of the 2023 high temperature anomaly, discussing the role of shipping emission sulphur reductions.

The paper is novel, topical and makes interesting findings on the short-term climate response to policy implementation.

I have questions over the quantification of the response as it seems very much on the high-side and this is not discussed. Also, for the paper to be of wider use its findings need to be contextualised better. The results also need to be better presented as it was hard for me to work out what they did or what the results meant, so the average reader might struggle to make sense of the results.

Nevertheless, I strongly support publication of this work after changes to better contextualise and explain their results.

Major points

1. Lines 32-41. From their figure 1 as radiative forcing of order 0.2 Wm-2 is stated, supported by three references. None of the references refer to the 2020 IMO regulation. Hodnebrog et al. does not even mention shipping emissions I believe. So the authors need to explain where their 0.2Wm-2 estimate come from. Other published work, not referenced, estimate forcing values of 0.2 Wm-2 over oceans – which is around 0.12 Wm-2 globally (e.g. Yuan et al., 2024; Forster et al., 2024). The authors also, and most importantly, diagnose their 0.2Wm-2 forcing estimate from their figure 1a's absorbed solar radiation. However, eyeballing the figure it looks to be at least 0.4 Wm-2. I think these forcing values need to be properly quantified. As if their forcing is closer to 0.4Wm-2, this would go a long way in explaining their high surface temperature response. Ideally, they should run an ensemble with prescribed SST and sea-ice to make a proper estimate of effective radiative forcing.

2. Fig 2 is quantifies the temperature response to shipping and is crucial but I found Section 2.2 did not do a good job describing this figure or its quantitative results and it took me as a reader a long time to work out what the figure meant. Panels 2 a) and b) don't seem to be referred to. The green lines seem the important ones to compare to the orange to estimate the role of shipping but these are not referred to or discussed. The figure is clearer than the text in my view. I think the section should be rewritten to walk the reader through the figure and quantitatively estimate the temperature change from shipping emission changes.

3. The paper would generally benefit a discussion about the high forcing estimate its uncertainty and how this may flow through into their high temperature response. This temperature response should also be put in the context of the other causes of high 2023 temperatures, such as the El Nino. The El Nino discussion as written, makes it look as if El Nino is not important – it may not be

important for modulating the shipping response, but otherwise it was. Also Forster et al. 2024, suggest that the aerosol radiative forcing, actually became more negative overall due to Canadian wildfires. In their global estimates, this more than compensated for shipping emission changes. Forster et al. may not be right, but this provides important context. This should be mentioned here.

4. The methods could be clearer – were the emission changes applied globally or over the ocean. I think comparing a 10 member ensemble to a 100-member control warrants explicit discussion as well. I notice that the green lines are more variable in Figures 1,2 and 3. This obviously affects the overall results and the uncertainty of the attribution, but how?

Other comments

5. Line 87. Planned non-CO2 policies do affect carbon budgets to a certain extent as non-CO2 warming scenarios are factored into carbon budget estimates. I think your statement needs more explanation.
6. Line 95, I think it's not only policy we need to worry about but possible human and natural emission changes in general?

---

## Author Comment (AC1)

Reviewer's comments are in black. Authors responses are in blue. *Changes to the text are in italic.*

This paper presents a useful contribution to the attribution of the 2023 high temperature anomaly, discussing the role of shipping emission sulphur reductions.

The paper is novel, topical and makes interesting findings on the short-term climate response to policy implementation.

I have questions over the quantification of the response as it seems very much on the high-side and this is not discussed. Also, for the paper to be of wider use its findings need to be contextualised better. The results also need to be better presented as it was hard for me to work out what they did or what the results meant, so the average reader might struggle to make sense of the results.

Nevertheless, I strongly support publication of this work after changes to better contextualise and explain their results.

We thank prof. Forster for his kind words and supportive comments. We address all of his comments below.

Major points

1. Lines 32-41. From their figure1 as radiative forcing of order 0.2 W m-2 is stated, supported by three references. None of the references refer to the 2020 IMO regulation. Hodnebrog et al. does not even mention shipping emissions I believe. So the authors need to explain where their 0.2 Wm-2 estimate come from. Other published work, not referenced, estimate forcing values of 0.2 Wm-2 over oceans – which is around 0.12 Wm-2 globally (e.g. Yuan et al., 2024; Forster et al., 2024). The authors also, and most importantly, diagnose their 0.2 Wm-2 forcing estimate from their figure 1a's absorbed solar radiation. However, eyeballing the figure it looks to be at least 0.4 Wm-2. I think these forcing values need to be properly quantified. As if their forcing is closer to 0.4Wm-2, this would go a long way in explaining their high surface temperature response. Ideally, they should run an ensemble with prescribed SST and sea-ice to make a proper estimate of effective radiative forcing.

   In our simulations the NET radiative forcing is of the order of 0.14 Wm-2 (please note this was corrected from the 0.2 Wm-2 previous value, due to an internal error in our calculation and estimate of the difference between two different ensembles of different sizes), in reference to fig. 1c, and as the reviewer pointed out the shortwave component (fig. 1a) is about 0.4 Wm-2. We acknowledge that

using simulations with prescribed sea surface temperatures (SST) and sea ice would be necessary for ERF calculation. However, to detect the aerosol signal within natural variability in the same context as the observational one, we decided to utilize the same LENS2 method.

Partanen et al. (2013) estimates an ERF of 0.37 (from -0.43 to -0.06) for a scenario with global shipping emission roughly corresponding to international agreements in 2020. We agree thatHodnebrog et al. (2024) are not appropriate references and we added a range of RF estimates from the suggested papers and other papers published recently:

*"The global NET RF effect as diagnosed by the change between LENS2 and NOSHIP (Fig. 1c, orange and green lines, respectively), is of the order of 0.14 W/m, within the range of previous estimates (with a minimum of +0.06 to a maximum of +0.37 W/m2 across different studies (Partanen et al., 2013; Yuan et al., 2024; Yoshioka et al., 2024; Skeie et al., 2024; Forster et al., 2024))."*

2. Fig2 quantifies the temperature response to shipping and is crucial but I found Section 2.2 did not do a good job describing this figure or its quantitative results and it took me as a reader a long time to work out what the figure meant. Panels 2 a) and b) don't seem to be referred to. The green lines seem the important ones to compare to the orange to estimate the role of shipping but these are not referred to or discussed. The figure is clearer than the text in my view. I think the section should be rewritten to walk the reader through the figure and quantitatively estimate the temperature change from shipping emission changes.

We changed the second paragraph in section 2.2 as follows:

*"Here we use our two ensembles to perform an attribution of the 2023 temperature impact by comparing their results in the three years following the change in emission between the two scenarios. We compare our model's results with the anomaly in one surface air temperatures dataset (Berkeley or measured anomaly), looking at monthly de-trended global temperature anomalies over the period 2020-2023, i.e. removing the, assumed linear, contribution from greenhouse gasses and seasonality. Figure 2b shows the monthly de-trending method applied to August months which has been used for each month of the 3-year time series in Figure 2a. For the NOSHIP ensemble we considered an average and high estimates of shipping emission impact (light and dark green lines, respectively), calculated as the ensemble average and ensemble average plus one standard deviation in NOSHIP. The likelihood of occurring temperatures in Figure 2c and d is calculated for the measured temperature anomalies (black line) and for the measured temperature anomalies without the average and high*

*contribution estimated from ship emissions (light and dark red lines corresponding to the light and dark green lines of panel 2a, respectively)."*

3. The paper would generally benefit a discussion about the high forcing estimate its uncertainty and how this may flow through into their high temperature response. This temperature response should also be put in the context of the other causes of high 2023 temperatures, such as the El Nino. The El Nino discussion as written, makes it look as if El Nino is not important – it may not be important for modulating the shipping response, but otherwise it was. Also Forster et al. 2024, suggest that the aerosol radiative forcing, actually became more negative overall due to Canadian wildfires. In their global estimates, this more than compensated for shipping emission changes. Forster et al. may not be right, but this provides important context. This should be mentioned here.

Practically all of the other studies on this subject in 2024 came out after we submitted ours, but we have now added a discussion of their results in the conclusions.

*"Our forcing estimates of +0.14 W/m2 ± 0.07 from CESM2 is located within the range of other works, which however use different methodologies to come to their conclusions. For instance, Yuan et al. (2024) found a forcing of 0.2 ± 0.11 W/m2 (over the global oceans) indirectly estimating it from cloud changes as simulated by NASA's Global Earth Observing System which would result in a global forcing very close to ours overall. However, their temperature estimate of 0.16K is twice as large as our estimate of 0.08K ± 0.03. Other studies like Skeie et al. (2024) tried to estimate the effective radiative forcing by conducting fixed-SSTs similar to ours, using four models (CESM2-CAM6, NASA GISS ModelE, NorESM2, OsloCTM3), and finding a range of 0.06 to 0.09 W/m2, similar to Yoshioka et al. (2024) which found 0.13 W/m2 using HadGEM3-GC3.1, very close to the 0.14 W/m2 ± 0.02 W/m2 found in UKESM by Jordan and Henry (2024) under similar experimental protocols. In this latter case, they also estimate the temperature response in UKESM to be 0.046K ± 0.010K. Finally, it was reported in Forster et al. (2024) that Gettelman and Yuan (2024) found 0.12 W/m2 using the FaIR climate emulator. A future assessment of the different methodologies used will be necessary to reconcile these estimates, perhaps coupling it with a rigorous multi-model assessment in CMIP7. A cause for our estimate being towards the higher end of others might be our use of fully coupled simulations, which may result in a warming-driven feedback on cloud forcing, and which we pursued to try to reconcile our estimates of both forcing and temperature changes with available observations. It is also possible that our*

*results are overestimated due to an excessive sensitivity of CESM2 to cloud-aerosol interaction, or that they are driven by our ensemble size: if this is the case, however, it will be necessary to find other explanations for the 2023 anomalous temperatures that currently don't seem to exist - especially as the persistent anomaly even after 12 months appears to rule out a statistical fluctuation."*

We also agree the sentence on ENSO contribution might be misleading and rephrased the sentence as follows:

*"Analysis of the ENSO state (Fig. S7) suggests that the anomaly cannot be attributed solely to a strong El Niño event starting in 2023, nor that the shipping emission change itself produced or magnified an El Niño event, as similar anomalies also appear in ensemble members during El Niña states."*

We also added a phrase in the Conclusions citing the Forster et al. 2024 paper, whose final publication was after our submission of this piece and so we missed in our literature review, but that clearly should be mentioned now.

*"Forster et al. (2024) also suggested that the 2024 global aerosol radiative forcing was made more negative due to the contribution from the Canadian wildfire. While our study is not suited to directly quantify such a potential contribution, due to prescribed biomass burning emissions that predate it, we note that our analyses of CERES fluxes show a global positive increase in Clear sky fluxes between 2023 and 2024 (Fig. A2) that is hard to reconcile with such a hypothesis. While it is certainly possible that increased warming resulted in higher wildfire risks at high latitudes (Fig. 3a), satellite data doesn't seem to support a forcing compensation between sulfate from shipping and wildfire aerosols. However, future studies including also realistic biomass burning could better clarify such matters".*

4. The methods could be clearer–were the emission changes applied globally or over the ocean. I think comparing a 10 member ensemble to a 100-member control warrants explicit discussion as well. I notice that the green lines are more variable in Figures 1,2 and 3. This obviously affects the overall results and the uncertainty of the attribution, but how?

We specified that emissions are reduced everywhere over the oceans:

*"[...] reducing sulfur emissions from shipping by 90 % everywhere over the oceans in accordance with IMO 2020 regulations from 2020 to 2030 […]"*

We specified also in methods the effect of the ensemble and we added the following reference at the end of the section:
*"Due to our computational constraints, we only performed 10 ensemble members compared to the original ensemble of 50. However, following Tebaldi et al. (2021) and Frankcombe et al. (2018), our ensemble size is representative of the perturbed state to allow us to estimate forcing changes "*

Other comments

5. Line87. Planned non-CO2 policies do affect carbon budgets to a certain extent as non-CO2 warming scenarios are factored into carbon budget estimates. I think your statement needs more explanation.

   We have changed the phrase to: "*For instance, it would be legitimate to ask if such policies should more explicitly be framed in terms of estimates of the remaining carbon budget before they're enacted?*"

6. Line95,I think it's not only policy we need to worry about but possible human and natural emission changes in general?

   Agreed. We have amended the phrase to add "*, and more broadly other possible human and natural emission changes,*"

---

## Author Comment (AC2)

Reviewer's comments are in black. Authors responses are in blue. *Changes to the text are in italic.*

The authors provide a new perspective on the contribution of reduced shipping sulfate emissions to the anomalous observed warming in 2023. While short-term climate forcers are mostly from continental emission sources, this work focuses on a maritime source. Further explanation is needed to make the attribution analysis more comprehensive and accessible to a broader audience. I have several concerns I would like the authors to consider:

We thank the reviewer for their comments, which we address below.

1. Temperature response to shipping emission reduction contributes considerably to observed temperature anomaly in 2023 (Fig. 2). Given that the CMIP6 SSP 3-7.0 scenario has been suggested to fail to capture East and South Asian anthropogenic aerosol emissions since 2006 (Zhang et al., 2019; Ramachandran et al., 2020), and has shown to have both local and remote impact (Wang et al., 2021; Xie et al., 2023). CESM2 LENS driven by SSP 3-7.0 underestimates the observed both net short-wave radiative flux and net short-wave minus long-wave radiative flux at the top of atmosphere since 2015 (Fig. 1) and, however, CESM2 LENS seems to capture the observed temperature anomaly in 2022 in Fig.2. Considering the low bias in radiative flux at the top of atmosphere in CESM2 LENS, it suggests other factors, such as internal variability, also contribute. Additionally, Fig. 3 shows anomalous warming in the tropical Pacific resembling an El Niño pattern. I recommend the authors discuss the impact of bias in emission scenarios and the potential impact of other factors.

   Our discussion has been updated, given also the input from reviewer 1. We have added some comments in Section 2.1 related to the useful references the reviewer provided: "*Our results are also consistent with previous studies in which aerosol emissions have already been shown to present a source of bias for CESM2 compared to observations (Zhang et al., 2019; Ramachandran et al., 2020) in other regions.*"

2. The authors estimate an increase in radiative forcing of 0.2 W/m² due to reduced shipping sulfate emissions, and aerosol-cloud interaction plays an important role. This is a significant magnitude, and I would expect a strong surface temperature response. There is no further explanation on how cloud response indirectly contributes to change in radiative forcing. The authors conclude that there is a three-year lag in surface temperature response due to ocean. I recommend showing results starting from 2020 and including the range of ensemble members in Fig 2. Additionally, a spatial map showing the geographic pattern of surface temperature

response, as well as full-sky radiative forcing, would be helpful, especially as shipping reductions are most pronounced in the North Atlantic.

In order to clarify the reviewer's point, we updated Figure A2 to highlight the difference between the forcing in All and Clear sky condition: we believe these analyses, in both CESM2 and CERES, depict clearly the contribution deriving from cloud-induced changes. In Fig. 3, we already show surface temperatures and the main component of radiative forcing of interest (in that case, we integrate absorbed solar radiation over the 3 years as a measure of absorbed energy, which more closely ties to oceanic temperatures). We have now added in the supplementary material (new Figure A3) also a map of aerosol and cloud visible aerosol optical depth anomalies in 2023 (note the difference in scale), and the cumulative SW cloud flux and net radiation anomalies, which we believe better support Figure 3.

For his other point, we note that we showed results starting in 2020 for both radiative forcing (Fig. 1) and global temperatures (Fig. A3), but we think Fig. 2 is clearer as is. The aim of the figure is to show the likelihood of occurring temperature based on observations with and without the impact of shipping emission estimated from the average and higher temperature anomalies from NOSHIP (that we calculate as the average and the average plus one standard deviation on the ensemble) and not showing the single member contribution with respect to the observations. We elaborated more about the method in the main text to make it clearer.

[Figure]

New Fig. A2: Time series of annual mean deviation from the 2000-2007 period for globally averaged (a) Absorbed Solar Radiation (ASR, defined as incoming minus outgoing shortwave), (b) Outgoing Longwave Radiation (OLR) and (c) ASR minus OLR (NET) radiative flux at the top of the atmosphere (TOA), in all sky (solid lines) and clear sky (dotted lines) conditions. The shaded area for CESM2-LENS2 simulations (Historical, SSP3-7.0, NOSHIP) represents one standard deviation calculated on ensemble members.

[Figure]

New Fig. A3: Maps of changes in sulfate and cloud visible optical depth in 2023 (a and b, respectively), and cumulative shortwave cloud flux and NET radiation over 2021-2023 (c and d, respectively) due to reduction in shipping emissions in CESM2-LENS2. Shaded areas indicate regions where the differences are not statistically significant at the 10% level, green contours indicate regions where the differences are not statistically significant at the 5% level.

I recommend including one or two more observational datasets or reanalyses to account for uncertainty. As ERA5 has shown bias in capturing observed top-of-atmosphere radiative fluxes, it is unclear why its results are included in the main figure and Fig. A5. The authors should clarify this choice. Additionally, the description of the Berkeley dataset is missing. Providing a brief description in the main text would be beneficial.

We added two more temperature datasets, the Global Surface Temperature (NOAAGlobalTemp) and the Met Office Hadley Centre/Climatic Research Unit global surface Temperature (HadCRUT5), which are now in the new Figures A4 and A5. Between January 2015 and May 2024 the new two datasets lie very closely between the lower estimate from Berkeley (black solid line) and higher estimate values from ERA5 (dark brown solid line), as shown in the revised Figure A5, and therefore they do not change our conclusions.

We believe our addition of ERA5 is useful here in comparison to observational datasets, especially as the temperature estimates are quite close to Berkeley and other measurements whereas the forcing presents such a large bias, pointing to the need to better include aerosol representation in reanalyses, which we think fits with

the main point of our paper. We did explore the use of other reanalysis products for radiative forcing, but we didn't find any others that performed better than ERA5, which we also chose based on its widespread use.

We now include a section about the various datasets in the *Methods appendix,* under *Observations and reanalysis*:

*"Sulfur shipping emissions are from the Community Emission Data System (CEDS) which provides estimates of emissions of anthropogenic greenhouse gases, reactive gases and aerosols, from 1750 to nowadays, based on existing emission inventories, emission factors, and activity/driver data (Hoesly and Smith, 2024).*

*To compare simulated radiative fluxes, we used satellite data from the Clouds and the Earth's Radiant Energy System (CERES) Energy Balanced and Filled Top-of-Atmosphere fluxes version 4.2 (CERES_EBAF_Edition4.2, NASA/LARC/SD/ASDC (2023)) and climate reanalysis data from the fifth generation European Centre for Medium-range Weather Forecasts (ECMWF) reanalysis (ERA5, Hersbach et al.).*

*Simulated surface temperatures are compared with ERA5, Berkeley Earth, Met Office Hadley Centre/Climatic Research Unit global surface temperature data set version 5.0.2.0 (HadCRUT5), and the NOAA Global Surface Temperature Dataset version 6.0 (NOAAGlobalTempv6). For the results in our Figure 2 we tested all datasets but ultimately only showed Berkley, as our conclusions were largely independent of the dataset chosen Fig. A5.*

*Berkeley Earth Land/Ocean Temperature Record (Rohde and Hausfather, 2020) combines the Berkeley Earth land-surface temperature field with an interpolated version of the Met Office Hadley Centre Sea Surface Temperature dataset version 4.0.0.0 (HadSST4).*

*HadCRUT5 (Morice et al., 2021) uses a statistical infilling method to integrate sea-surface temperature data from the HadSST4 with land-surface air temperature data from the Climatic Research Unit temperature dataset version 5.0.0.0 (CRUTEM5) .*

*NOAAGlobalTempv6 (Huang and Zhang) combines the land-ocean surface temperature analysis from the Extended Reconstructed Sea Surface Temperature (ERSSTv5) with land surface air temperature analysis, which are is on the Global Historical Climatology Network-Monthly (GHCN-M) temperature database."*

[Figure]

New Fig A4: Time series of 12-months rolling mean of global mean temperature changes from 1981-2000 for CESM2-LENS2 simulations (Historical, SSP3-7.0, NOSHIP) and observations and reanalysis (Berkeley, ERA5, NOAAGlobalTempv6, HadCRUT5). The shaded area for CESM2-LENS2 simulations represents one standard deviation calculated on ensemble members.

[Figure]

New Fig A5: Time series of detrended global mean temperature changes from the 1981-2000, distinguished for each month of the year, for CESM2-LENS2 simulations (Historical, SSP3-7.0, NOSHIP) and observations and reanalysis (Berkeley, ERA5, NOAAGlobalTempv5, HadCRUT5). The shaded area for CESM2-LENS2 simulations represents one standard deviation calculated on ensemble members.

Reference

Ramachandran, S., Rupakheti, M. and Lawrence, M.G., 2020. Aerosol-induced atmospheric heating rate decreases over South and East Asia as a result of changing content and composition. Scientific Reports, 10(1), p.20091.

Zhang, Q., Zheng, Y., Tong, D., Shao, M., Wang, S., Zhang, Y., Xu, X., Wang, J., He, H., Liu, W. and Ding, Y., 2019. Drivers of improved PM2. 5 air quality in China from 2013 to 2017. Proceedings of the National Academy of Sciences, 116(49), pp.24463-24469.

Xie, Y., Huang, J., Wu, G., Lei, N. and Liu, Y., 2023. Enhanced Asian warming increases Arctic amplification. Environmental Research Letters, 18(3), p.034041.

Wang, Z., Lin, L., Xu, Y., Che, H., Zhang, X., Zhang, H., Dong, W., Wang, C., Gui, K. and Xie, B., 2021. Incorrect Asian aerosols affecting the attribution and projection of regional climate change in CMIP6 models. npj Climate and Atmospheric Science, 4(1), p.2.